# AMIC—Autologous Matrix-Induced Chondrogenesis Technique in Patellar Cartilage Defects Treatment: A Retrospective Study with a Mid-Term Follow-Up

**DOI:** 10.3390/jcm9041184

**Published:** 2020-04-20

**Authors:** Daniele Tradati, Paola De Luca, Alessio Maione, Francesco Mattia Uboldi, Piero Volpi, Laura de Girolamo, Massimo Berruto

**Affiliations:** 1U.O.S Chirurgia Articolare del Ginocchio, I Clinica Ortopedica, ASST Gaetano Pini-CTO Gaetano Pini, Piazza Cardinal Ferrari 1, 20122 Milan, Italy; tradati@outlook.com (D.T.); alessio.maione@gmail.com (A.M.); fraubo@gmail.com (F.M.U.); massimo.berruto@fastwebnet.it (M.B.); 2IRCCS Istituto Ortopedico Galeazzi, Orthopaedic Biotechnology Lab, 20161 Milan, Italy; deluca.paola@grupposandonato.it; 3U.O. Ortopedia del ginocchio e Traumatologia dello sport, Istituto Clinico Humanitas IRCCS, Via Alessandro Manzoni 56, 20089 Rozzano, Italy; piero.volpi@humanitas.it

**Keywords:** patellar cartilage defects, Autologous Matrix Induced Chondrogenesis, retrospective cohort study, cartilage repair

## Abstract

Background: Knee cartilage defects can be retrieved in 60% of patients undergoing knee arthroscopy, especially in the patellofemoral joint. Different techniques have been proposed to treat patellar defects, although most of them are associated with short-term results. In this study Autologous Matrix Induced Chondrogenesis (AMIC), combining subchondral microfractures with a collagen membrane (type I and III collagen), was used in the treatment of isolated patellar cartilage defects. Methods: Twenty-four patients were enrolled in this retrospective study. Subjective-International Knee Documentation Committee (IKDC), Visual Analog Scale for Pain (VAS), and Kujala score were collected at 1, 3, 6, and 12 months after surgery, whereas the Tegner Activity Level Scale was determined preoperatively and at final follow-up (final-FU). The same postoperative management and rehabilitation protocol was adopted for all the patients. Results: Fourteen patients met the inclusion–exclusion criteria and were evaluated at a mean final-FU of 68.2 months (range 25.4–111.2). At 12 months, Kujala, IKDC, and VAS scores significantly increased in comparison to the preoperative assessment, whereas no statistically significant differences were reported between 12 months and final follow-up. Conclusion: This study demonstrated very good results throughout the follow-up, also in sports patients. The AMIC technique, together with an adequate rehabilitation protocol, can be considered as a reliable one-step alternative for the treatment of large isolated patellar cartilage defects.

## 1. Introduction

Cartilage defects are a common condition often leading to further joint degeneration into osteoarthritis that represents a considerable social and economic burden for society. Chondral lesions could be retrieved in up to 60% of knee arthroscopies and in 19% of them a grade IV chondromalacia could be identified, mainly located on the patellar cartilage [1]. In a retrospective study reporting 2931 patients with cartilage lesions, the patellar articular surface was involved in 37.5% of the patients [2]. These data were confirmed by a systematic review on the prevalence of chondral defects in athletes’ knees which reported that 37% of the defects were located on the patellofemoral joint (patella and trochlea) [3].

The chondral defects of patella can be related to different causes such as abnormal patellofemoral pressure, due to the lateral compression of the patella over the trochlea [4,5], direct trauma, acute patellar dislocation, or being the result of excessive overload secondary to malalignment and/or maltracking. Patella fracture or osteochondritis dissecans can also be included among possible causes [6].

Usually patellar chondral defects are associated with severe pain, swelling, and disability that can result in progressive degeneration into osteoarthritis, even in younger patients [7]. Regenerative techniques demonstrated better clinical and objective results in patients with femoral condyle lesions in comparison to patellar ones, probably because of the higher thickness of the patellar cartilage [8].

Although the presence of tissue specific progenitor cells in the pathological articular cartilage, potentially able to promote a regenerative process, has been recently confirmed [9], they are not able to start this process in the absence of proper biomechanical stimuli.

Different surgical approaches have been developed and adapted to treat patellar defects, including bone marrow stimulation techniques, chondral/osteochondral transplantation, both autologous or allogenic, mosaicplasty, and autologous chondrocyte implantation (ACI) [7].

Microfracture is a minimally invasive and cheap technique that provides an effective short-term functional improvement of knee function, but mid-term data have demonstrated a progressive worsening of the outcomes [8,10]. The limitations of this technique rely on the variable repaired cartilage volume, the limited hyaline-quality of the repaired tissue, the lack of physical and chemical bonding between macromolecular components of the repair tissue, and the residual adjacent cartilage [10]. 

One of the reasons for the limited quality of the repair tissue could be ascribed to the poor number of mesenchymal progenitor cells deriving from microfractures and the high probability to have these cells washed away from the lesion site if non-protected [11]. To overcome these limitations, different membranes or matrices have been proposed to both protect blood clotting and to provide a mechanical and biological support to cells. They have been developed using different biomaterials, such as collagen, collagen-hydroxyapatite, hyaluronic acid, and chitosan [12]. In order to further increase the quality of the regenerated tissue, some authors proposed an improvement to the outcome of this technique by adding healing promoters like Platelet Rich Plasma (PRP) [13], concentrated bone marrow aspirate [14], and cells [15].

Among the matrix-assisted techniques, the most recognized and studied is the AMIC (Autologous Matrix Induced Chondrogenesis) procedure, combining microfractures with a collagen I/III membrane (Chondro Gide; GeistlichPharm, Wolhusen, Switzerland). This one stage procedure, indicated for the treatment of grade III or IV cartilage lesions according to the ICRS (International Cartilage Repair Society) classification [16], ranged between 2 and 8 cm^2^ [7,17]. Satisfactory results, in terms of pain reduction and functional improvement, have been already reported for up to 7 years [14,18,19,20,21,22,23,24]. A very recent meta-analysis concerning the clinical evidence on the use of AMIC in the knee confirmed that it can be recommended for the treatment of knee chondral and osteochondral defects [25]. In particular, good outcomes were also demonstrated in patients presenting patellofemoral cartilage defects although at a short follow-up [26,27].

The aim of this retrospective study was to evaluate the medium-term outcome of patients with patellar lesions who underwent an AMIC procedure at an average 5.6 years follow-up. 

## 2. Experimental Section

### 2.1. Study Design and Patient Selection

In this retrospective study, a consecutive series of patients who underwent cartilage repair of patella by using an open AMIC technique was considered. Patients were eligible for inclusion in the study if they (1) had an isolated patellar lesion ICRS stage > 3A, (2) were younger than 50 years at the time of the surgery, (3) had patellar AMIC procedure, and (4) had a follow-up longer than 24 months. The following exclusion criteria were applied: (1) associated or previous patellar cartilage surgery, (2) varus or valgus greater than 5°, (3) previous tibial or femoral intra-articular fracture, (4) patellar osteoarthritis (OA) (Iwano grade 3–4) [28], tibiofemoral OA (Kellgren–Lawrence grade 3–4), (5) trochlear kissing lesions or (6) trochlear dysplasia (grade B high-level or D according to the Dejour classification) [29].

### 2.2. Data Collection and Ethical Approval

Preoperative and intraoperative findings were extracted from the internal institutional database. Age at the time of surgery, sex, comorbidities, associated surgical procedures, lesion(s)’s location, size and staging according to the ICRS classification were collected and recorded. All the patients were evaluated routinely before surgery and at 1, 3, 6, 12 months after the surgical procedure. At the time of the study (final-FU) patients were re-contacted and a further assessment was performed. At each time point subjective- International Knee Documentation Committee (IKDC) [30], Visual Analog Scale for Pain (VAS) [31], and Kujala score [32] were recorded. A patient’s activity level was determined preoperatively and at final follow-up using the Tegner Activity Level Scale [33]. The overall patient satisfaction according to the Insall and Crosby grading system was also collected at the final time point [34].

The study was designed based on the criteria of the Declaration of Helsinki and approved by an external Ethics Committee (ASL Città di Milano, 21/07 MS, protocol n° 471/07). Advantages and disadvantages of the procedure were explained to the patients and an informed consent was obtained before their inclusion in the study.

### 2.3. AMIC: Patient Evaluation, Indications, and Surgical Technique

The surgical indication to the AMIC procedure relied on anamnestic, clinical, and radiological findings. The patient’s clinical history was collected in order to assess previous traumatic events, fractures, patellar dislocation, osteochondritis dissecans (OCD) or knee surgeries. Anterior knee pain referred to as a bar beneath the patellar apex, exacerbated by stair climbing, sitting position, and squatting, was observed in the majority of the patients included in this study. All the patients experienced persistent pain notwithstanding previous conservative treatment. Exercises focused on isometric quadricep strengthening, external-rotators reinforcement, and stretching of both hamstring and extensor mechanism were encouraged for at least 12 months before indicating surgical repair. 

Every patient underwent a full clinical examination preoperatively, in order to exclude patellar instability, ligament lesions, hyperlaxity (greater than 7 according to the Beighton score [35]), and other common causes of anterior knee pain. A focal tenderness at the palpation of the articular patellar cartilage was a common finding in these patients. 

X-ray evaluation including full weight-bearing long radiograph, standard knee in full weight-bearing, skyline and Rosenberg–Schuss view was performed to exclude osteoarthritis, varus-valgus deformities, severe trochlear dysplasia (type B high grade or D according to Dejour classification) patella alta and patella baja (according to the Caton–Dechamp Index). MRI allowed the estimation of the approximate size of chondral lesions and the possible involvement of the subchondral bone. 

Indication to the AMIC procedure was therefore given in the case of anterior knee pain not responding to at least 12 months of conservative treatment, and chondral–osteochondral lesion (area ≥2.5 cm^2^, depth ≥50% of the surrounding healthy cartilage profile) detected by MRI. Patients were not considered indicated to the procedure in the cases of (1) uncorrected lower limb malalignment (>5° varus or valgus knee), (2) uncorrected patellar instability factors (patella height, tibial tubercle trochlear groove distance, high degrees trochlear dysplasia) [36], (3) patellofemoral OA (Iwano grade 3–4), (4) tibiofemoral OA (Kellgren–Lawrence grade 3–4), (5) unstable knee, (6) neoplastic, autoimmune or metabolic diseases, (6) previous knee localized infections, (7) diffuse patellar bone edema. 

At the time of the surgery, the patient was positioned supine and the table was set to allow 90° of knee flexion. The tourniquet was placed tight and activated at the beginning of the arthrotomic procedure. A diagnostic arthroscopy was performed in order to confirm the chondral patellar lesion and evaluate the overall status of the knee joint. An 8–12 cm medial parapatellar skin incision was then performed with the knee at 90° of flexion. A standard medial parapatellar capsulotomy allowed the articular surfaces of the patellofemoral joint to be exposed.

The knee was placed in full extension and the kneecap was partially everted trying to minimize the trauma of the surrounding soft tissues (Figure 1). 

The chondral lesion was measured and debrided till the calcified layer was visible, taking care to preserve stable and perpendicular shoulders. A template corresponding to the size of the prepared lesion was created by using an aluminum foil (Figure 2).

The microfractures of the subchondral bone were performed by a dedicated awl, from the periphery to the center of the defect, paying attention to maintain a distance of 3–4 mm between each hole (Figure 3). 

Lastly, the collagen matrix, properly shaped using the template, was placed on the defect and fixed with fibrin glue (Figure 4 and Figure 5).

The tourniquet was then released to allow proper bleeding of the subchondral bone through the microfractures. 

The stability of the implant was tested by several cycles of flexion–extension, temporarily fixing the capsule with a surgical clamp. A drainage without aspiration was placed intra-articularly, whereas the capsule and the other tissue were sutured following the standard way.

When needed, additional procedures, such as medial patellofemoral ligament (MPFL) reconstruction and lateral release, were performed after the suture of the capsule. 

### 2.4. Postoperative Management and Rehabilitation Protocol

Immediately after surgery the knee was placed in a full-extension locked brace and quadricep isometric exercises were initiated from the first postoperative day. The drainage was removed between 24 and 48 h after the procedure, depending on the knee status. After 48 h a gentle progressive mobilization of the knee was started by using a continuous passive motion device. Progressive partial weight-bearing with crutches, with brace in full extension, was allowed starting from the second postoperative day. The primary endpoint of the rehabilitation protocol was defined as maintaining complete extension and reaching 90° of flexion in the first 4 weeks. Crutches and brace were removed at 6 weeks after surgery if a good quadriceps control was achieved. Water-assisted exercises were encouraged as soon as the wound was completely healed, whereas squats and the stationary bicycle were gradually introduced after 3 months from the procedure. 

A more intensive rehabilitation protocol was introduced 6 months after surgery, including isometric quadriceps reinforcement with weights and elastic bands and advanced cardiovascular fitness exercises.

Return to sport was allowed after 9–12 months if the patient was asymptomatic and the strength of the thigh muscle was completely restored (90% of the contralateral limb in isokinetic tests).

### 2.5. Statistical Analysis

All the analyses were performed using SPSS software for Windows (v.20.0). The Kolmogorov–Smirnov test was used in order to verify the normal distribution of the following continuous variables: age at surgery, area of the lesion, preoperative Kujala score, preoperative IKDC score, preoperative VAS score, and preoperative Tegner activity level. A two-way repeated measures ANOVA test with Bonferroni correction was performed in order to assess statistically significant differences in terms of IKDC, VAS, and Kujala scores at each follow-up. “Time” was considered as within-subject factor and “sex” was considered as between-subjects factor. An independent sample t-test was used in order to compare preoperative and postoperative Tegner activity level. Statistical significance was achieved if *p* < 0.05.

## 3. Results

Among the 24 patients considered for this study, 14 met the inclusion–exclusion criteria (9 males, 5 females). The mean age at surgery was 38.4 ± 10.4 years while the mean area of the lesion was 4.5 ± 0.7 cm^2^. According to the ICRS classification, a grade 3A lesion was reported in 3 patients, grade 3B and 4 lesions in 5 and 6 patients, respectively. The mean follow-up was 68.2 ± 31.8 months. No surgery-related adverse events were recorded during the entire duration of the follow-up period. Twelve patients underwent isolated patellar AMIC procedure, whereas in 2 cases the patients required associated surgeries due to the concomitant patellar instability: 1 open lateral release and 1 MPFL reconstruction. 

Age at surgery, preoperative Kujala score, preoperative IKDC score, and Tegner Activity scale resulted as being normally distributed, while preoperative VAS and the area of the lesion were not. 

With respect to the preoperative values, at 12 months of follow-up Kujala score passed from 49.6 to 87.8 (*p* < 0.01), the IKDC score from 36.1 to 79.8 (*p* < 0.01), and the VAS score from 7.5 to 1.5 (*p* < 0.01). No statistically significant difference was reported between 12 months and final follow-up evaluation (Figure 6a–c). Subgroup analysis showed no statistically significant difference between males and females considering both IKDC and Kujala score; nevertheless females demonstrated an increase in VAS score between 12 months and final follow-up (from 1.7 to 4.0).

Preoperative Tegner Activity Level increased from 1.9 to 4.3 (*p* < 0.01) at final follow-up (Figure 6d).

Patient’s satisfaction was rated good or excellent in 22% and 78% of patients, respectively (Figure 7).

None of the patients needed additional procedures at the target knee during the follow-up period.

## 4. Discussion

The main finding of the study is that the AMIC technique enabled satisfactory and stable outcomes for the treatment of isolated cartilage patellar defects at a mean follow-up of 5.6 years.

To the best of our knowledge, the current study represents one of the largest case series of AMIC patellar cartilage repairs with the longest mean follow-up available in the literature. 

The AMIC technique is a one-step procedure for the treatment of knee cartilage defects. Many authors have reported good or excellent clinical results at mid-term follow-up in the management of knee chondral defects [25].

Focusing on patellar cartilage lesions, the literature presents just a few studies, most of which include only a limited number of patients. A study reports the results at 2 years follow-up of 10 patients with patellofemoral cartilage defects who underwent arthrotomic AMIC procedure [1]. The mean Kujala score improved from 41.9 to 59.8 at final follow-up and VAS score decreased from 7.39 to 3.94. Nevertheless, the limited follow-up and the inclusion of trochlear lesions make it difficult to extrapolate conclusions about the long-term results of isolated patellar cartilage defects. In the present study only patellar cartilage defects were considered, fibrin glue fixation was preferred instead of sutures, and microfractures of the patellar were performed instead of microdrilling. Although having comparable preoperative Kujala scores (41.9 vs. 49.6), in the present study higher postoperative values were reported (59.8 vs. 87.8) albeit for the longer follow-up. Surgical technique and patient selection could have been relevant factors influencing the outcome, such as microdrilling, which in the authors’ opinion could have compromised the subchondral bone microvascularization therefore leading to lower clinical results. 

Additionally, Kusano et al. [27] obtained good and comparable results with the present study. The authors treated 20 patients with full-thickness chondral defects of the patella with a mean follow-up of 29.3 ± 2.3 months demonstrating, with respect to preoperative values, an improvement of IKDC (from 51 to 74) and a decrease of VAS (from 6 to 2). Nevertheless, no improvement was observed in term of the Tegner score. In this case, also the follow-up is quite short and 90% of these patients included in the analysis underwent associated patellar unloading osteotomy or realignment procedure and lateral release. Moreover, nine patients over 20 underwent mobilization under anesthesia due to postoperative knee stiffness.

More recently, Sadlik et al. [37] reported the results of 12 patients with isolated patellar cartilage lesions treated with an arthroscopic AMIC technique at a mean follow-up of 38 months. Mean IKDC score improved from 37.4 to 90.1 at final follow-up and VAS score decreased from 7.8 to 2.3, with no failure reported.

The choice of the authors to adopt a full arthroscopic approach, including patients with no extensor mechanism abnormalities, could explain the higher values of postoperative IKDC score in comparison to the present study. Nevertheless, the arthroscopic technique could be technically more demanding and not suitable for patients with fixed lateral patellar subluxation with lateral patellar facet defects.

The postoperative protocol is crucial in order to optimize the surgical outcome and not to undermine the repair process. The rehabilitation protocol should not only be focused on function restoration but also on promoting and sustaining the tissue regenerative processes [38,39]. In ideal conditions, the physical and biochemical environment should promote cell differentiation towards an articular cartilage-like profile, leading to the development of a durable and organized tissue [39,40,41,42]. After surgery, mechanical stimuli (flexion, rotation, translation) play a primary role in promoting the development and maintenance of the regenerated tissue, activating surface receptors, integrins, the ions channel, and adhesion complexes [43,44,45]. According to Schätti et al. [46], compression and shear forces alone could not properly induce chondrogenesis, while the application of shear, superimposed upon dynamic compression, could significantly increase gene expression leading to a higher concentration of glycosaminoglycan and collagen II fibers.

According to this evidence, in the current study an early passive mobilization was allowed 48 h after the surgery, aiming to provide early low-level shear stress stimulation.

Nevertheless, to avoid potential harmful stress over to the collagen scaffold, the patients were allowed to walk with full weight-bearing only by using a full-extension brace, and knee flexion was limited to 90° for the first 30 postoperative days, since deeper flexion was reported to increase patellofemoral combined shear and compression forces [47]. These specific elements heavily characterize the patellofemoral joint rehabilitation protocol, in comparison to the one adopted for the femoral and tibial lesions, and this fact should be strongly noted in order to allow an optimal outcome. 

Another key point in the late postoperative treatment is the return to sport activity. Kreuz et al. emphasized the correlation between continuous sport participation and clinical long-term outcomes after cartilage repair procedures [48]. In our study, sport activity was encouraged between 9 and 12 months postoperatively, in patients presenting a near-full quadricep recovery and no pain. At a mean follow-up of 68.2 months, mean Tegner Activity Scale was 4.3 while 36% of patients reported a score equal or greater to five. Based on this evidence, return to sport should not be discouraged in patients who underwent patellar AMIC procedure, already at mid-term term follow-up. In this regard, the return to sport could be facilitated by augmenting the matrix-assisted microfracture technique with biologics, the so-called AMIC plus procedure. A pilot study including five patients with patellar cartilage defects treated with AMIC procedure combined with PRP, reported an improvement in both clinical score and subjective knee pain in comparison to the preoperative condition. The mean improvement in Kujala score was 31 points, whereas in a previous study of the same author using AMIC technique alone the mean improvement was 18 points. However, the limited follow-up (24 months) and number of patients do not allow any conclusive results to be drawn about the use of this enhanced technique in the patellofemoral joint. A recent randomized prospective study comparing the long-term outcomes after AMIC and AMIC + bone marrow aspirate concentrate, in the treatment of focal knee chondral lesions showed that AMIC+ allows for faster recovery from injury [14]. This would indicate the addition of a biological product for patients requiring a prompt return to activity. Similarly, medium-term results in a small series have demonstrated durable clinical and imaging improvement when the matrix-assisted microfracture technique was associated with bone marrow concentrate in the treatment of critical-size knee chondral lesions [49], as well as in the talus [50]. Further evolutions of the AMIC technique could lead to even better and longer-lasting results, but additional studies are necessary to establish a significant clinical improvement, especially in relation to the patellofemoral joint.

Among the limitations of this study we acknowledge its retrospective nature, the lack of a control group, the unavailability of MRI at final follow-up not permitting a radiological evaluation of the repair procedure to be performed, and the limited number of patients. In one patient, the size of the lesion t the arthroscopic evaluation was 1.0 cm^2^ a. Nevertheless, the quality of the surrounding cartilage at the probing test as well as the concomitant massive bone marrow lesion at the MRI evaluation, led the surgeon to proceed with the AMIC procedure. The correctness of this choice was confirmed during the arthrotomic debridement after which the lesion area was greater than the 2 cm^2^.

## 5. Conclusions

In the current study good to excellent results at a mean follow-up of 5.6 years were found in patients with isolated patellar chondral lesions treated with the Autologous Matrix-Induced Chondrogenesis (AMIC) technique. Female gender was associated with increased VAS scores at the last score at final follow-up in comparison to the 12 month evaluation. 

The AMIC technique could be considered as a reliable one-step procedure in the treatment of patients with large isolated patellar cartilage defects, highly symptomatic and recalcitrant to conservative treatment. Sports return should not be discouraged in sport-active patients, but not before 9–12 months after surgery and then following a specific rehabilitation protocol.

## Figures and Tables

**Figure 1 jcm-09-01184-f001:**
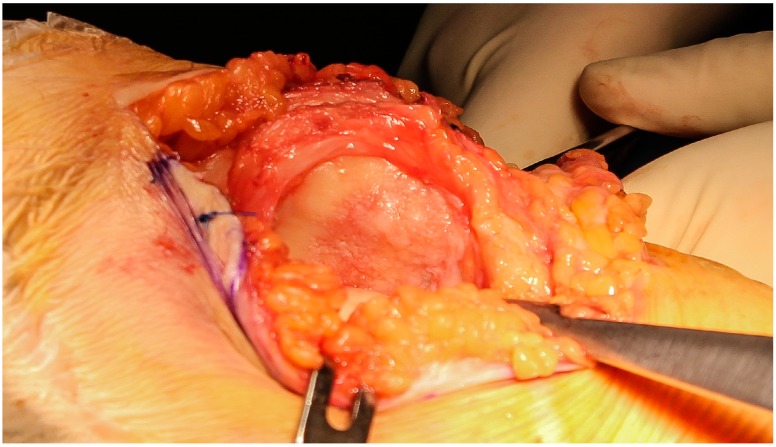
Medial-parapatellar approach to expose the chondral lesion.

**Figure 2 jcm-09-01184-f002:**
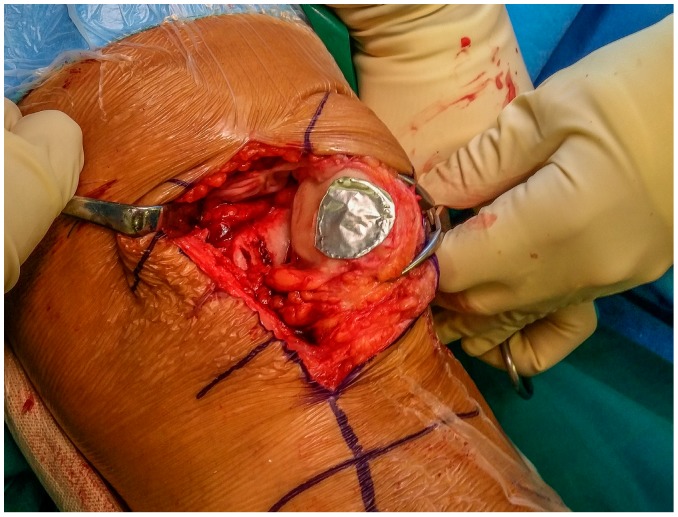
Preparation of the template corresponding to the size and shape of the chondral.

**Figure 3 jcm-09-01184-f003:**
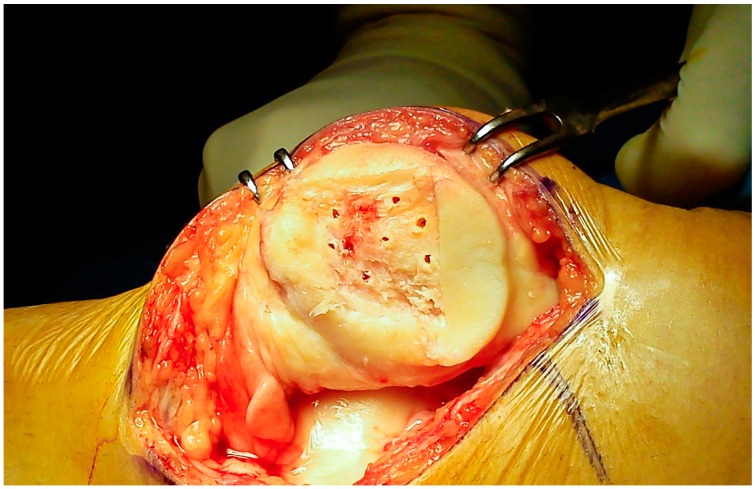
Microfractures of the subchondral bone. A distance of 3–4 mm was maintained between the holes to avoid microfractures confluence.

**Figure 4 jcm-09-01184-f004:**
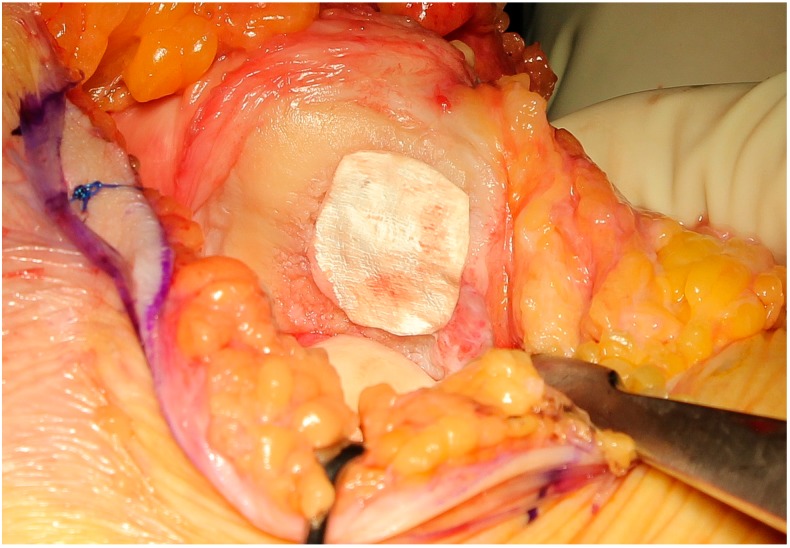
Positioning of the collagen matrix over the cartilage defect, avoiding oversizing.

**Figure 5 jcm-09-01184-f005:**
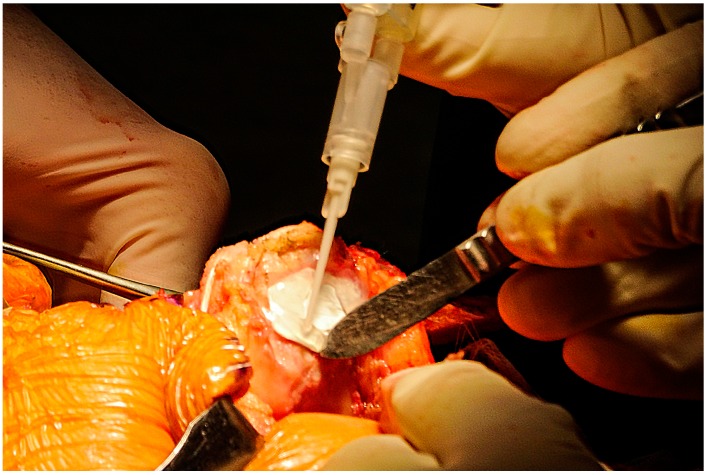
Sealing of the matric by the fibrin glue.

**Figure 6 jcm-09-01184-f006:**
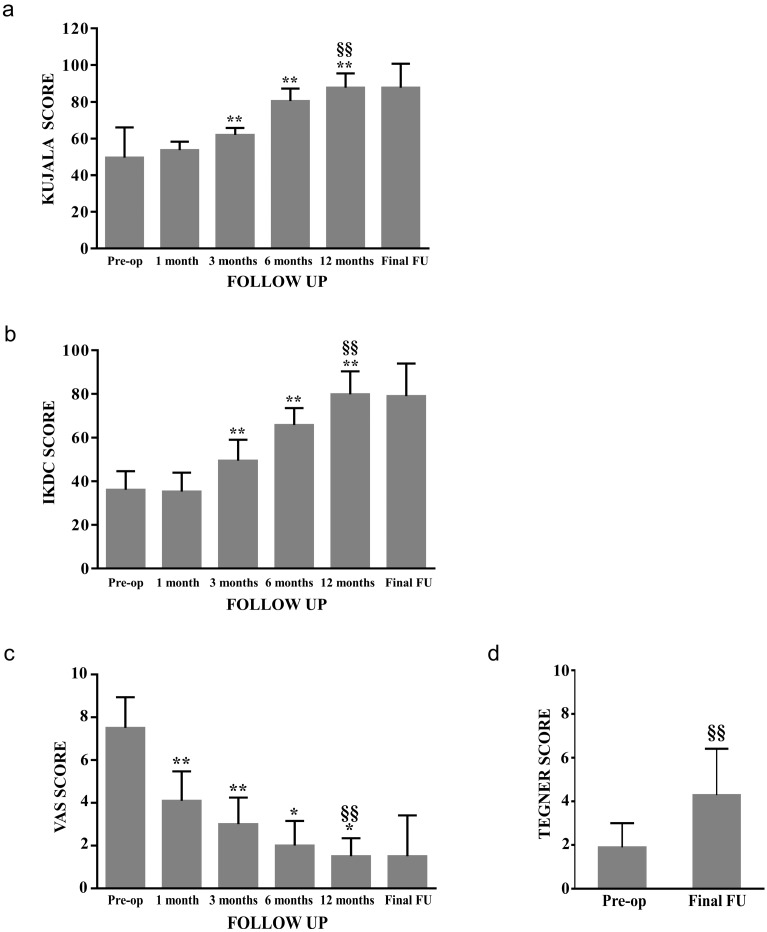
Clinical score, pain trend and activity level after patellar Autologous Matrix Induced Chondrogenesis (AMIC) procedure. Kujala (**a**), International Knee Documentation Committee (IKDC) (**b**), Visual Analog Scale for Pain (VAS) (**c**) and Tegner Activity Level Scale (**d**). * *p* < 0.05 and ** *p* < 0.01 vs. previous value. ^§§^
*p* < 0.01 vs. preoperative value.

**Figure 7 jcm-09-01184-f007:**
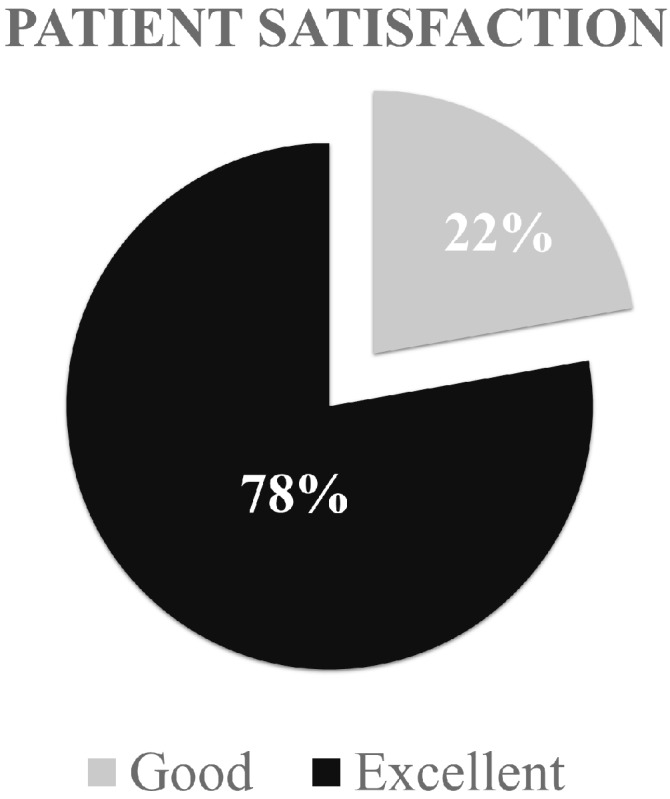
Patient satisfaction rate according to the Insall–Crosby grading system.

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
