# Peer review of "AMIC—Autologous Matrix-Induced Chondrogenesis Technique in Patellar Cartilage Defects Treatment: A Retrospective Study with a Mid-Term Follow-Up"

_jcm, 2020, doi:10.3390/jcm9041184_

Round 1
Reviewer 1 Report
I have found the material interesting, but in my opinion, something needs to be modified:
The authors describe the results as "long term follow-up". However, for cartilage repair it is usual to consider the follow-ups at 5-6 years as “medium term”, while the follow-ups over 7-8 years are considered “long term”. I recommend changing this aspect both in the title and in the manuscript.
In the abstract, type of study (retrospective) and final follow-up (68.2 months) are not specified. They have to added it to the abstract.
Multiple studies evaluated the utility and efficacy of various approaches to overcome the limitations of the microfracture technique. I recommend an overview in the introduction of other microfracture enhancing techniques which are also promising (e.g. other scaffolds with or without cells, biologics augmentations).
In the discussion, the authors should describe the possibility of new options to improve AMIC technique also in patellar cartilage defects, such as “AMIC plus”, already used in chondral lesions of femoral condyles and in osteochondral lesion of the talus.
There are some typing errors throughout the manuscript that need to be corrected:
- 86: “Kellgren_Lawrence” Use en dash (-) in “Kellgren_Lawrence”
- 96: The reference 31 is not correct and not related to Insall and Crosby grading system. Change it.
- 164-165: Provide the same decimal numbers for both mean and standard deviation / range.
- 254: Add the dot at the end of the sentence.
Author Response
Response to Reviewer 1 Comments
- The authors describe the results as "long term follow-up". However, for cartilage repair it is usual to consider the follow-ups at 5-6 years as “medium term”, while the follow-ups over 7-8 years are considered “long term”. I recommend changing this aspect both in the title and in the manuscript.
- We thank the reviewer for this note. The title as well as the manuscript were modified accordingly
- In the abstract, type of study (retrospective) and final follow-up (68.2 months) are not specified. They have to added it to the abstract.
- The abstract was updated according to the request (lines 23 and 28)
- Multiple studies evaluated the utility and efficacy of various approaches to overcome the limitations of the microfracture technique. I recommend an overview in the introduction of other microfracture enhancing techniques which are also promising (e.g. other scaffolds with or without cells, biologics augmentations).
- We thank the reviewer for this note. While we acknowledge the importance of the existing literature about other microfracture enhancing technique, we had decided not to mention them in detail to respect the required length of manuscript and more in particular of the introduction. However, we have now included a brief description of the most represented biomaterials as well as the biological augmentations to these techniques accompanied by relevant citations (line 72-76)
- In the discussion, the authors should describe the possibility of new options to improve AMIC technique also in patellar cartilage defects, such as “AMIC plus”, already used in chondral lesions of femoral condyles and in osteochondral lesion of the talus.
- We thank the reviewer to rise this comment. The possible options to improve AMIC have been now included in the discussion (Line 290-307)
- There are some typing errors throughout the manuscript that need to be corrected:
- 86: “Kellgren_Lawrence” Use en dash (-) in “Kellgren_Lawrence”
- Amended
- 96: The reference 31 is not correct and not related to Insall and Crosby grading system.
- Amended
- 164-165: Provide the same decimal numbers for both mean and standard deviation / range.
- Decimals were properly adjusted
- 254: Add the dot at the end of the sentence.
- Added
- 86: “Kellgren_Lawrence” Use en dash (-) in “Kellgren_Lawrence”
Best regards
Reviewer 2 Report
The manuscript identifies that autologous matrix-induced chondrogenesis can be effectively used for the treatment of patients with large isolated patellar cartilage defects who are highly symptomatic and not responding to the conservative treatment. The design of experiments, analysis, and follow-up experiments are appropriately carried out to conclude the study. The study does provide new clinical knowledge to the field of cartilage repair and regeneration although there are very few studies that have reported the results of patients with patellar lesions treated with an arthroscopic AMIC. However, the manuscript in the present form has poor readability in terms of language and requires extensive editing.
Specific comments:
There is a lack of information particularly in the materials and methods section, specifically concerning the recruitment of patients. Is there any clinicopathological information about the patients included in the study? The posthoc test employed to assess the significance of the repeated measures two-way ANOVA should be mentioned. Was the ethical approval obtained for the study? The name of the institution that issued the approval should be listed.
Author Response
Response to Reviewer 2 Comments
- The manuscript in the present form has poor readability in terms of language and requires extensive editing.
- The manuscript was revised from a native English speaker
- There is a lack of information particularly in the materials and methods section, specifically concerning the recruitment of patients. Is there any clinicopathological information about the patients included in the study?
- We thank to reviewer for this note. Patient Evaluation and surgical indications have been now added (line 114-140)
- The posthoc test employed to assess the significance of the repeated measures two-way ANOVA should be mentioned.
- The Post-hoc test (Bonferroni correction) has been now included in the statistical analysis paragraph (line 195)
- Was the ethical approval obtained for the study? The name of the institution that issued the approval should be listed.
- The study obtained the approval of the Ethics Committee. All the details were already reported in the original submission (now line 109-110).
Best regards